# Periprosthetic Joint Infection Caused by Gram-Positive Versus Gram-Negative Bacteria: Lipopolysaccharide, but not Lipoteichoic Acid, Exerts Adverse Osteoclast-Mediated Effects on the Bone

**DOI:** 10.3390/jcm8091289

**Published:** 2019-08-23

**Authors:** Mei-Feng Chen, Chih-Hsiang Chang, Chih-Chien Hu, Ying-Yu Wu, Yuhan Chang, Steve W. N. Ueng

**Affiliations:** 1Bone and Joint Research Center, Chang Gung Memorial Hospital, Taoyuan 33305, Taiwan; 2Department of Orthopedic Surgery, Chang Gung Memorial Hospital, Taoyuan 33305, Taiwan; 3College of Medicine, Chang Gung University, Taoyuan 33305, Taiwan; 4Graduate Institute of Clinical Medical Sciences, College of Medicine, Chang Gung University, Taoyuan 33305, Taiwan

**Keywords:** periprosthetic joint infection, lipopolysaccharide, lipoteichoic acid, bone mineral density, osteoclasts, trabeculae, bone loss, aseptic loosening, reimplantation, arthroplasty

## Abstract

Periprosthetic joint infection (PJI)—the most common cause of knee arthroplasty failure—may result from Gram-positive (GP) or Gram-negative (GN) bacterial infections. The question as to whether PJI due to GP or GN bacteria can lead to different rates of aseptic loosening after reimplantation remains open. We have investigated this issue through a retrospective review of clinical records obtained from 320 patients with bacterial PJI. The results revealed that, compared with GP infections, GN infections were associated with an increased risk of aseptic loosening. In animal studies, mice underwent intrafemoral injection of lipopolysaccharide (LPS) from GN bacteria or lipoteichoic acid (LTA) from GP bacteria. We demonstrate that LPS—but not LTA—reduced both the number of trabeculae and the bone mineral density in mice. In addition, LPS-treated mice exhibited a reduced body weight, higher serum osteocalcin levels, and an increased number of osteoclasts. LPS accelerated monocyte differentiation into osteoclast-like cells, whereas LTA did not. Finally, ibudilast—a toll-like receptor (TLR)-4 antagonist—was found to inhibit LPS-induced bone loss and osteoclast activation in mice. Taken together, our data indicate that PJI caused by GN bacteria portends a higher risk of aseptic loosening after reimplantation, mainly because of LPS-mediated effects on osteoclast differentiation.

## 1. Introduction

Periprosthetic joint infection (PJI)—the most common cause of knee arthroplasty failure—accounts for 16–25% of failed total knee replacements [1,2,3] and is the third most common indication for revision hip arthroplasty [4,5,6,7,8]. There is also evidence that, compared with patients with aseptic loosening, those with PJI may have a higher re-revision rate (21% versus 4.3%, respectively) [9,10,11,12], although the exact pathophysiological mechanisms underlying this phenomenon remain elusive. It is possible that certain bacterial components may elicit a persistent proinflammatory response, even after antibiotic-induced bacterial death, ultimately resulting in bone resorption [13]. The activation of toll-like receptors (TLRs) has been implicated in this process [14]. Recent animal studies have also shown that bacterial osteomyelitis delays fracture fixation [15]. Although accumulating evidence supports the detrimental effects of bacteria on bone formation [16], the question as to whether PJI caused by Gram-positive (GP) or Gram-negative (GN) bacteria can lead to different rates of aseptic loosening after reimplantation remains open.

Lipopolysaccharide (LPS) and lipoteichoic acid (LTA)—the main cell wall components of GN and GP bacteria, respectively—have been shown to affect bone formation in experimental studies [14,17]. LPS has inhibitory effects on osteoblasts by inhibiting their differentiation [18] and promoting apoptosis [19]. It can also decrease alkaline phosphatase activity and downregulate osteogenic genes [20], ultimately inducing inflammatory bone loss [21]. Interestingly, exposure to GP bacteria-derived LTA inhibits the differentiation of bone marrow-derived macrophages to osteoclasts [22]. It is thus possible that, while LPS inhibits osteogenesis [23], the opposite effects may be elicited by LTA. In an effort to further clarify whether GP and GN bacterial infections may exert different effects on the bone, we designed the current study with three specific aims. First, we performed a clinical study to examine whether PJI caused by GP or GN bacteria can lead to different rates of aseptic loosening after reimplantation. Second, we conducted animal experiments in which the differential effects of the intrafemoral injection of LPS and LTA were examined, with a special focus on osteoblasts and osteoclasts [24,25]. Finally, we used a preclinical model to test whether ibudilast—a toll-like receptor (TLR)-4 antagonist—may prevent bacterial-induced bone resorption.

## 2. Patients and Methods

This retrospective study was approved by our institutional review board and was compliant with current ethical standards (IRB Number: 201800763B0C501). We performed a single-center retrospective review of clinical records. In 2018, we began to analyze the clinical records of 531 patients who presented with PJI (274 hips and 257 knees) between 2011 and 2016. All patients with PJI were scheduled to undergo two-stage exchange arthroplasty. The definition of PJI followed the Musculoskeletal Infection Society (MSIS) criteria in keeping with previous methodology [26]. Among the 531 patients, we identified 320 cases of bacterial PJI (160 hips and 160 knees). The microbiological examinations conducted in the 211 excluded patients revealed that PJI was either culture-negative or caused by infections from anaerobes, fungi, or mycobacteria. After a minimum follow-up of 2 years, 79 patients required re-revision. Reasons for re-intervention were classified as aseptic loosening versus other causes (infections, intensity instability, or fractures). All cases of loosening were found to be aseptic, according to the MSIS criteria.

### 2.1. Experimental Animal Studies

All animal procedures complied with the National Institute of Health guidelines and were reviewed and approved by the local Hospital Animal Care and Use Committee (IACUC approval number 2016082901). Ten-week-old male C57BL/6 mice were purchased from BioLasco Biotechnology (Taipei, Taiwan). Animals were initially anesthetized through an intraperitoneal injection of a 0.01 mL/kg mixture (1:1 volume/volume) of tiletamine-zolazepam (Zoletil^®^; Virbac, Carros, France) and xylazine hydrochloride (Rompun^®^; Bayer HealthCare AG, Leverkusen, Germany), and subsequently subjected to intrafemoral injections of 10 mg/kg LPS (from *Escherichia coli* O127:B8; Sigma-Aldrich, St. Louis, MO, USA) or 20 mg/kg LTA (from *Staphylococcus aureus*; Sigma-Aldrich) in phosphate-buffered saline (PBS; 10 μL). Seven to ten days post-injection, mice were sacrificed (5–7 mice per group). The inoculated femur underwent immediate 10% formaldehyde fixation and was subjected to micro-CT analysis.

### 2.2. Ibudilast Treatment

Following the intefemoral delivery of LPS or LTA, mice received daily intraperitoneal injections (4 mg/kg/day) of either ibudilast (Cayman Chemical, Ann Arbor, MI, USA) or a vehicle (PBS) for three consecutive days. The in vitro effects of ibudilast (10 μM) were tested by adding the drug to cell culture media.

### 2.3. Serum Osteocalcin Assay

Serum osteocalcin levels were measured using a commercially available ELISA kit (BioSource Europe SA, Nivelles, Belgium), according to the manufacturer’s protocol.

### 2.4. Micro-Computed Tomography Bone Imaging

Nondestructive ultrastructural bone analysis was performed with a SkyScan 1176 micro-CT scanner (Bruker, Kontich, Belgium). Samples were wrapped into saline-soaked gauze and subsequently scanned using a 0.5 mm aluminum filter with the following parameters: voltage, 60 kV; current, 417 μA; and exposure time, 1000 ms. Images were reconstructed to a 9 μm pixel resolution using the GUP-NRecon software (version 1.7.4.2; Bruker, Kontich, Belgium) and analyzed with the Skyscan CTAn program (version 1.15.4.0, Bruker, Kontich, Belgium). Regions of interest were defined as trabecular bone areas of 1−3 mm^2^ located below the growth plate (231 slices). Trabecular bone was automatically isolated with the CTAn software, which was also used to calculate both morphometric indices and the bone mineral density (BMD). The density reference was validated using BMD calibration phantoms with two different hydroxyapatite concentrations (0.25 and 0.75 g/cm^3^, respectively). Illustrative three-dimensional images were obtained with the Skyscan CTVox program (version 3.3.0, Bruker, Kontich, Belgium).

### 2.5. Histochemistry and Immunofluorescence Staining

Femur samples were harvested in neutral-buffered formalin (10%), incubated in a rapid decalcifier solution, trimmed, and paraffin-embedded. Four mm thick sections were subsequently stained with (1) hematoxylin/eosin, (2) Masson’ trichrome, and (3) tartrate-resistant acid phosphatase (TRAP; Takara, Shiga, Japan). The number of TRAP-positive multinucleated (≥3 nuclei) osteoclasts was quantified under a light microscope (DFC7000 T, Leica Microsystems, Wetzlar, Germany). Primary antibodies against cathepsin K (1:100, ab19027, Abcam) and osterix (1:100, ab22552, Abcam) were applied for immunofluorescent staining. Samples were subsequently incubated with secondary antibodies, including an Alexa Fluor^®^ 488-conjugated anti-rabbit IgG (1:200, A21206, Invitrogen, Carlsbad, CA, USA), for 60 min at 25 °C.

### 2.6. Cell Culture and Osteoclast Differentiation

RAW264.7 cells were plated on a three-well-chamber slide (Nunc Lab-Tek, ThermoFisher Scientific, Waltham, MA, USA; density: 1 × 10^4^ cells per well) and maintained in alpha-Eagle’s Minimum Essential Medium (MEM) medium containing 10% fetal bovine serum (FBS) and antibiotics. Cells were differentiated for five days into mature osteoclasts in the presence of 100 ng/mL LPS, 100 ng/mL LTA, recombinant receptor activator of nuclear factor kappa-Β ligand (RANKL, 50 ng/mL) alone, recombinant RANKL plus LPS, and recombinant RANKL plus LTA. The formation of mature osteoclasts was assessed with cathepsin K staining. Wheat germ agglutinin (WGA) was stained with the Alexa Fluor™ 594 conjugate (ThermoFisher Scientific), whereas 4’6-diamidino-2-phenylindole (DAPI) staining was used for nuclei (D1306; Invitrogen).

### 2.7. Statistical Analysis

All data were obtained from at least three independent experiments. Quantitative data are given as means ± standard errors of the mean and analyzed with two-way analysis of variance (ANOVA), followed by Bonferroni’s post-hoc tests. Body weight values were analyzed using two-way repeated-measures ANOVA followed by Tukey’s post-hoc test. Categorical variables were examined with Fisher’s exact test. Statistical calculations were performed with SPSS 22.0 (IBM, Armonk, NY, USA) and GraphPad Prism 7.0 (GraphPad Inc., San Diego, CA, USA). Two-tailed *p* values < 0.05 were considered statistically significant.

## 3. Results

### 3.1. PJI Caused by Gram-Negative Bacteria Increases the Risk of Aseptic Loosening

Of the 320 cases with bacterial PJI (160 hips and 160 knees), 79 patients required re-revision after a minimum follow-up of 2 years. Reasons for re-intervention were classified as aseptic loosening versus other causes (infections, instability, or fractures). The results revealed that, compared with GP infections, GN infections were associated with an increased risk of aseptic loosening (Table 1). Figure 1 depicts the Kaplan–Meier survival analysis of reoperation rates due to aseptic loosening in patients with periprosthetic joint infection caused by Gram-positive versus Gram-negative bacteria. The time free from reoperation was significantly lower in the GN-PJI group compared with the GP-PJI group (*p* < 0.001).

### 3.2. Intrafemoral Injection of LPS, but not LTA, Results in a Decreased Number of Trabeculae and a Lower Bone Density

The results of micro-CT 3D scanning revealed that an intrafemoral injection of LPS in mice reduced the number of bone trabeculae (Figure 2A)—an effect that was not observed upon the injection of LTA. Moreover, quantitative analysis of micro-CT data (Figure 2B) indicated that LPS, but not LTA, reduced numerous parameters of bone density, including the bone volume density (BS/TV), bone volume fraction (BV/TV), BMD, trabecular number (Tb.N), and trabecular thickness (Tb.Th). Five to seven mice were used in each experimental group.

### 3.3. LPS, But Not LTA, Increases the Number of Osteoclasts

Upon hematoxylin/eosin and Masson’s trichrome staining, bone tissues of LPS-treated mice showed marked loosening that resembled that of necrotic trabecular bone (Figure 2C)—a finding which did not occur in LTA-treated mice. Immunofluorescence staining was subsequently used to examine the activation of osteoblasts and osteoclasts under different experimental conditions. Osteoclast activity—examined by cathepsin K staining [27]—was significantly higher in LPS-treated mice, but not in those receiving an LTA injection (Figure 2D). Osterix staining, which reflected osteoblast activity, was unaffected by either treatment. Quantitative fluorescence intensity analysis of osteoblasts and osteoclasts at 7–10 post-injection days confirmed these findings (Figure 2E,F).

### 3.4. Intrafemoral Injection of LPS in Mice Decreases the Body Weight and Increases Serum Osteocalcin Concentrations

An intrafemoral injection of LPS, but not LTA, resulted in a significant decrease of body weight over time (Figure 2G). Compared with the baseline, body weight changes observed between day 1 and day 3 were 90/86/89% and 93/93/96% in LPS- and LTA-treated mice, respectively. Serum osteocalcin—a biomarker of bone turnover [28]—was found to increase significantly at day 10 post-injection in LPS-treated, but not in LTA-treated, mice (Figure 2H).

### 3.5. LPS, but not LTA, Promotes the Differentiation of Monocytes into Osteoclast-Like Cells

The results of in vitro staining experiments revealed that LPS, but not LTA, was able to promote monocyte differentiation into osteoclast-like cells (Figure 3A). The effect elicited by LPS was further exacerbated in the presence of RANKL (Figure 3B). Similar results were also observed with regard to the number of TRAP-positive multinucleated (≥3 nuclei) cells (Figure 3D)—a finding confirmed by quantitative analysis (Figure 3C,E). Altogether, these results indicate that LPS can promote osteoclast differentiation in vitro.

### 3.6. In Vitro Effects of Ibudilast on LPS-Induced Osteoclast Differentiation

We next conducted in vitro experiments to examine whether ibudilast—a TLR4 antagonist [29]—can inhibit LPS-induced osteoclast differentiation. Both staining and quantitative results revealed that ibudilast can effectively reduce the number of cathepsin K-positive multinucleated (≥3 nuclei) cells elicited by LPS (Figure 4A,B). Similar results were obtained for TRAP-positive multinucleated (≥3 nuclei) cells (Figure 4C,D). Altogether, these in vitro results indicate that ibudilast is capable of inhibiting LPS-induced osteoclast differentiation.

### 3.7. Ibudilast Attenuates LPS-Induced Femoral Bone Loss in Mice

Animal experiments were subsequently conducted to confirm that ibudilast can protect against LPS-induced osteolysis in vivo. Micro-CT analysis of mouse femoral bone revealed that ibudilast administration following LPS injection resulted in a higher number of bone trabeculae compared with positive control experiments conducted with LPS alone (Figure 5A). Similar results were obtained in terms of bone density-related indicators (Figure 5B).

### 3.8. Ibudilast Attenuates LPS-Induced Bone Loosening and Reduces the Number of Osteoclasts In Vivo

Hematoxylin/eosin and Masson’s trichrome staining revealed that the femoral bone obtained from mice subjected to LPS injection had a loose structure—a finding which was abrogated when LPS was followed by ibudilast administration (Figure 5C). Similar results were obtained when cathepsin K staining was used to quantify osteoclast activity (Figure 5D)—a finding subsequently confirmed by quantitative analysis of osteoclast fluorescence intensity (Figure 5E). The graphical abstract is shown in Figure 6.

## 4. Discussion

The overall goal of this study was to shed more light on the differential effects of GP and GN bacteria in the pathogenesis of PJI. We initially examined this issue in relation to the risk of aseptic loosening in a large clinical cohort of patients with PJI. The results revealed that, compared with GP infections, GN infections were associated with an increased risk of aseptic loosening. In order to shed more light on this phenomenon, we conducted a series of in vitro and in vivo experiments aimed at clarifying the distinctive effects on the bone of LPS and LTA—which derive from GN and GP bacteria, respectively. We found that LPS, but not LTA, reduced both the number of trabeculae and the bone mineral density. In addition, LPS-treated mice exhibited a reduced body weight, higher serum osteocalcin levels, and an increased number of osteoclasts. LPS was found to accelerate monocyte differentiation into osteoclast-like cells, whereas LTA did not. Finally, ibudilast—a toll-like receptor (TLR)-4 antagonist—was found to inhibit LPS-induced bone loss and osteoclast activation in mice.

The number of primary total knee and total hip arthroplasties is projected to markedly grow in the future due to the most common degenerative joint disease [30], an increase which will be accompanied by a parallel need for revisions [31]. Infection is a major cause of failure following revision total knee arthroplasty (44.1% of all cases) [32], whereas it plays a role in 6% of revision total hip arthroplasty [33,34,35]. Loss of the acetabular bone and pelvic discontinuity present challenges in the revision of total hip arthroplasty [36,37]. Moreover, the inability to eradicate infection in knee PJI may be as high as 33%, resulting in elevated complication rates [38]. Our data show, for the first time, that patients with PJI due to GN bacteria have a significantly higher risk of aseptic loosening than those caused by GP bacteria. The cellular mechanisms underlying this phenomenon were investigated in a series of in vitro and in vivo experiments for which we used key components of GN and GP bacteria (i.e., LPS and LTA, respectively). We selected specific cell wall components because there is evidence that dead bacteria can elicit a persistent proinflammatory response, even after successful antibiotic treatment [13]. Our experimental results indicated that, compared with LTA from GP bacteria, LPS from GN bacteria decreases the number of trabeculae, leads to a lower bone density, increases the number of osteoclasts and serum osteocalcin concentrations, and promotes the differentiation of monocytes into osteoclast-like cells. All of these detrimental effects elicited by LPS can explain the increased risk of aseptic loosening observed in patients with PJI caused by GP bacteria.

Published data indicate that dead bacteria mainly cause bone loss through immune reactions elicited by bacterial cell wall components, including lipopolysaccharide (LPS) and lipoteichoic acid (LTA). Endotoxin-adherent wear particles may contribute to the aseptic loosening of orthopedic implants, even in the absence of clinical or microbiological evidence of infection [39,40,41]. However, only a few studies have provided direct evidence that LPS can cause bone loss in vivo. Tatro et al. demonstrated that endotoxin wear particles on orthopedic implants may influence the rate of osteolysis, both in a murine calvaria model and in patients with aseptic loosening [42]. In a murine tooth extraction model, Tomomatsu et al. reported that the bone mineral density (BMD) of the tooth socket was significantly reduced by LPS injection [23]. Using LPS-doped polyethylene particles, Liu et al. investigated the effect of LPS on bone resorption in a rat model [43]. The results revealed that LPS-induced impaired fixation was directly associated with increased bone resorption and reduced bone formation, thus offering an explanation for the clinical occurrence of LPS-related implant osteolysis and loosening [43]. Bonsignore et al. demonstrated that LPS adherent to titanium alloy discs affected the later stages of osteogenic differentiation and mineralization in vitro [17]. These results indicate that bacterial debris may act as a surface contaminant that can impair the osseointegration of orthopedic implants. Abu-Ame et al. conducted a seminal study on the effect of LPS on osteoclasts and showed that LPS-induced osteoclastogenesis is mediated by the tumor necrosis factor and/or its p55 receptor [44]. To our knowledge, only one study has investigated the effect of lipopeptides on the bone mineral density in mice [45]. The results revealed that lipoproteins are a bacterial component that promotes osteoclast differentiation and activation [45]. Because the effects of LPS and LTA on bone formation have not been thoroughly studied, we sought to investigate their impact on bone formation and osseointegration.

LPS and LTA released from the cell wall of GN and GP bacteria, respectively, may exert their biological effects via interaction with TLRs [46]. Because TLR4 acts as a specific LPS receptor, we also examined the potential usefulness of ibudilast in the prevention of LPS-induced bone loss. Our preclinical results consistently indicated that this drug has the capacity to prevent or attenuate most of the LPS-associated detrimental effects on the bone. Albeit preliminary, these findings may open new therapeutic avenues related to the use of ibudilast for the prevention of aseptic loosening in patients with PJI caused by GP bacteria. Bone loss due to inflammatory reactions may be inhibited by pharmaceutical inhibition of the pathway, and our ibudilast results support the latest published research results [47,48,49,50].

TLR4 and TLR2 act as macrophage receptors for bacterial LPS and LTA, respectively. As a specific TLR4 antagonist, ibudilast does not affect TLR2 signaling and does not impair the ability of macrophages to recognize Gram-positive bacteria. Moreover, ibudilast does not inhibit phagocytosis of bacteria by neutrophils and does not affect the acquired immune response. Because aseptic loosening may be caused by LPS-induced excess activation of osteoclasts, coating implants with a TLR4 antagonist may have potential to prevent future aseptic loosening in patients with GN-PJI. This local preventative strategy may avoid adverse effects that may stem from systemic administrations of TLR4 antagonists.

Our findings need to be interpreted in the context of some limitations. First, this is a single-center experience and the results might not be generalizable. Second, we cannot be sure that the amount of LPS and LTA administered in animal experiments does actually reproduce a clinical situation.

In conclusion, the results of our study indicate that PJI caused by GN bacteria portends a higher risk of aseptic loosening after reimplantation, mainly because of the LPS-mediated effects on osteoclast differentiation. Our data may pave the way towards the development of new therapeutic strategies for PJI.

## Figures and Tables

**Figure 1 jcm-08-01289-f001:**
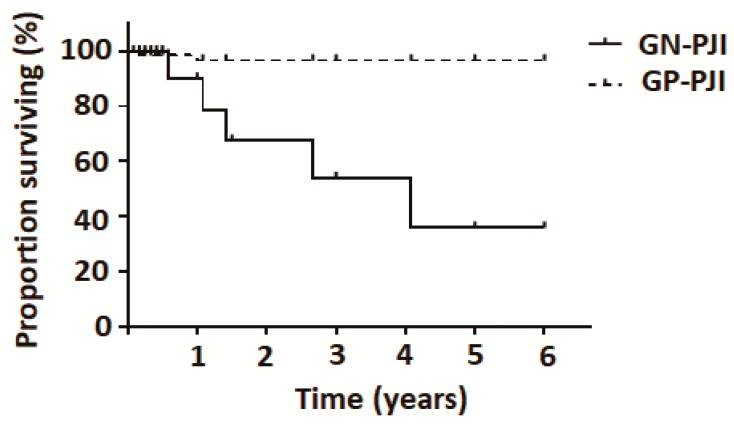
Kaplan–Meier plot of reoperation rates due to aseptic loosening in patients with periprosthetic joint infection caused by Gram-positive (*n* = 251) versus Gram-negative (*n* = 69) bacteria. The time free from reoperation was significantly lower in the GN-PJI group compared with the GP-PJI group (*p* < 0.001).

**Figure 2 jcm-08-01289-f002:**
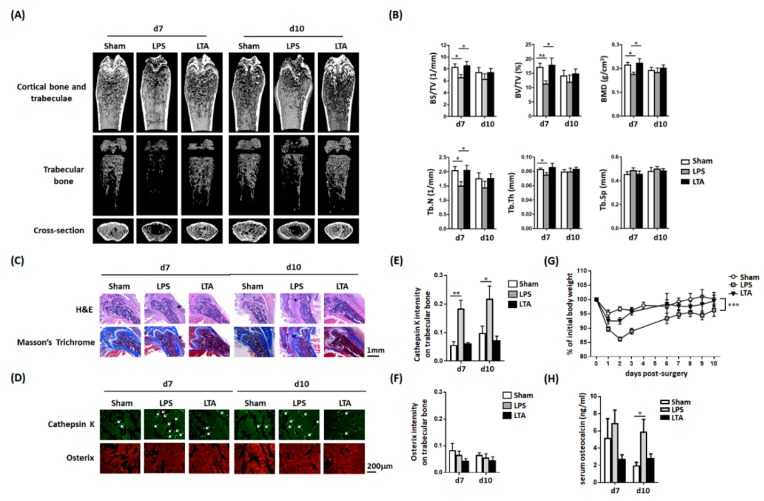
Lipopolysaccharide and lipoteichoic acid exert distinct effects on the bone mineral density, bone morphology, cathepsin K expression, and serum osteocalcin concentrations in mice. (**A**) Mice were subjected to an intrafemoral injection of LPS, LTA, or PBS (vehicle). The results of micro-CT revealed that LPS, but not LTA, decreased the number of bone trabeculae. (**B**) Quantitative results of micro-CT analysis in mice treated with PBS (*n* = 7), LPS (*n* = 5), or LTA (*n* = 5). LPS, but not LTA, was found to reduce morphometric bone indices. (**C**) Hematoxylin/eosin and Masson’s trichrome staining revealed a dense bone morphology in PBS-treated mice, whereas LPS injection resulted in bone loosening (black arrow)—a finding which was not observed in LTA-treated mice. (**D**) Immunofluorescence was used to detect cathepsin K (an osteoclast marker) and osterix (an osteoblast marker). LPS, but not LTA, increased the cathepsin K fluorescence intensity, whereas osterix expression was unaffected by either treatment. Quantitative expression of cathepsin K (**E**) and osterix (**F**) in mice treated with PBS, LPS, and LTA. (**G**) Body weight was measured daily in mice treated with PBS, LPS, and LTA and all results were normalized to the initial weight of each mouse. (**H**) Serum osteocalcin levels were measured by ELISA at indicated time points after injection. Data are presented as means ± standard errors of the mean. Analyses were conducted with two-way ANOVA, followed by Bonferroni’s post-hoc test. ** p* < 0.05, ** *p* < 0.01, *** *p* < 0.001. Abbreviations: d, day; LPS, lipopolysaccharide; LTA, lipoteichoic acid; BS, bone surface; BV, bone volume; TV, tissue volume; BMD, bone mineral density; Tb.N, trabecular number; Tb.Th, trabecular thickness; Tb.Sp, trabecular spacing; PBS, phosphate-buffered saline.

**Figure 3 jcm-08-01289-f003:**
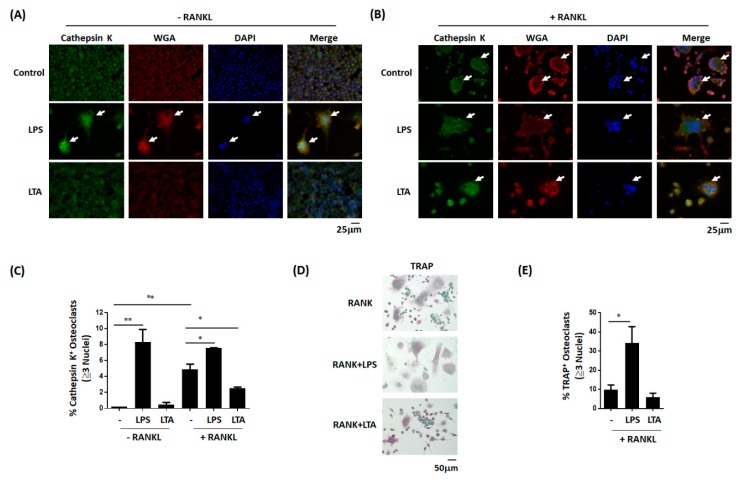
Lipopolysaccharide and lipoteichoic acid have a distinct effect on osteoclast differentiation and activation in vitro. (**A**) LPS promoted the differentiation of the murine monocytic cell line RAW264.7 into osteoclast-like cells—an effect which was not elicited by LTA. (**B**) LPS accelerated RANKL-induced differentiation of RAW264.7 cells into osteoclast-like cells. (**C**) These results were confirmed by quantitative data analysis. Data are presented as means ± standard errors of the mean. Analyses were conducted with two-way ANOVA, followed by Bonferroni’s post-hoc test. (**D**) LPS promoted the differentiation of RAW264.7 cells into TRAP-positive osteoclast-like cells. (**E**) Quantitative data analysis confirmed that LPS, but not LTA, accelerated RANKL-induced differentiation of TRAP-positive osteoclast-like cells. Data are presented as means ± standard errors of the mean. Analyses were conducted with unpaired Student’s *t*-tests. ** p* < *0.05*, ** *p* < 0.01. LPS, lipopolysaccharide; LTA, lipoteichoic acid; RANKL, recombinant receptor activator of nuclear factor kappa-Β ligand; TRAP, tartrate-resistant acid phosphatase.

**Figure 4 jcm-08-01289-f004:**
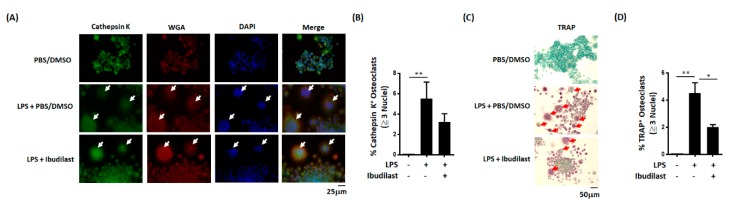
Ibudilast inhibits LPS-induced osteoclast differentiation in vitro. (**A**) Representative image illustrating the inhibitory effect of ibudilast on the differentiation of cathepsin K-positive osteoclasts. RAW264.7 cells were cultured for 5 days with LPS (100 ng/mL) in the presence of ibudilast (10 μM) and subsequently stained for cathepsin K. (**B**) Quantitative data analysis confirmed the inhibitory effect of ibudilast against the formation of cathepsin K-positive multinucleated (≥3 nuclei) cells in vitro. (**C**) Representative image illustrating the inhibitory effect of ibudilast on the differentiation of TRAP-positive osteoclasts. (**D**) Quantitative data analysis confirmed the inhibitory effect of ibudilast against the formation of TRAP-positive multinucleated (≥3 nuclei) cells in vitro. Data are presented as means ± standard errors of the mean. Analyses were conducted with unpaired Student’s *t*-tests. * *p* < 0.05, ** *p* < 0.01. LPS, lipopolysaccharide; TRAP, tartrate-resistant acid phosphatase.

**Figure 5 jcm-08-01289-f005:**
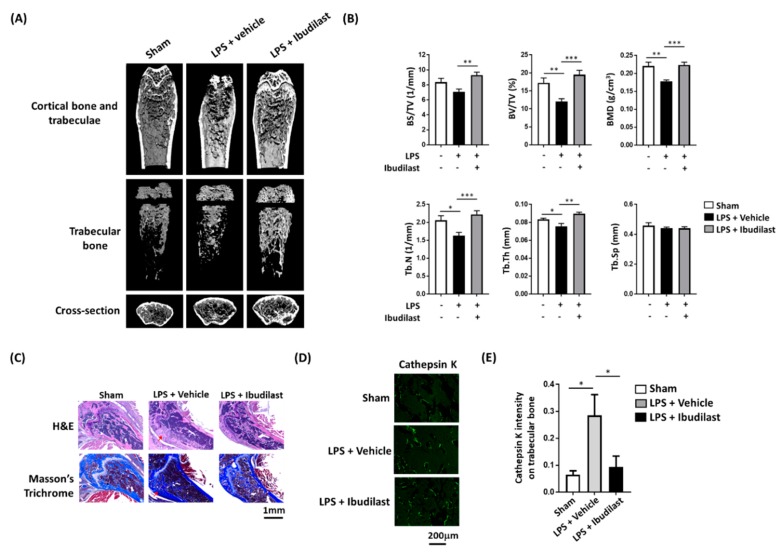
Intraperitoneal administration of ibudilast following intrafemoral injection of LPS protects mice against LPS-induced alteration in bone morphology and prevents osteoclast activation. (**A**) At one hour after intrafemoral LPS injection, mice received intraperitoneal ibudilast or PBS (vehicle) once daily for 3 consecutive days. Micro-CT images revealed that ibudilast exerts protective effects against LPS-induced reduction of bone density. (**B**) Quantitative data analysis confirmed these findings. (**C**) Hematoxylin/eosin and Masson’s trichrome staining revealed a dense bone morphology in PBS-treated mice, whereas LPS injection resulted in bone loosening (black arrow)—a finding which was attenuated by ibudilast administration. (**D**) Immunofluorescence was used to detect cathepsin K (an osteoclast marker). LPS increased cathepsin K fluorescence intensity—a finding which was attenuated by ibudilast administration. (**E**) Quantitative data analysis confirmed these findings. Data are presented as means ± standard errors of the mean. Analyses were conducted with two-way ANOVA followed by Tukey’s *post-hoc* test. ** p* < *0.05*, ** *p* < 0.01, *** *p* < 0.001. LPS, lipopolysaccharide; PBS, phosphate-buffered saline.

**Figure 6 jcm-08-01289-f006:**
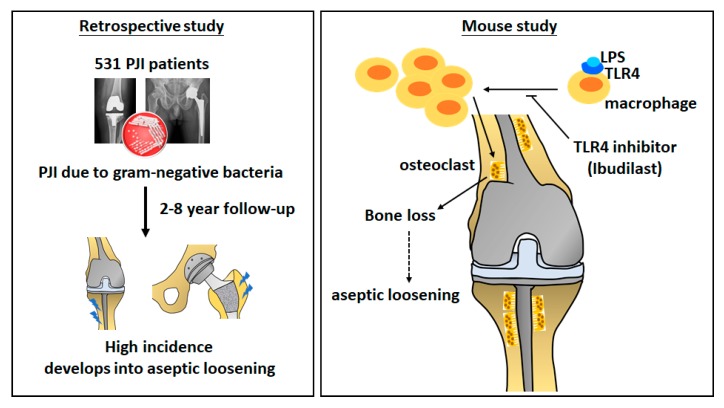
Graphical abstract. We initially conducted a retrospective review of clinical records obtained from 320 patients with bacterial PJI. The results revealed that, compared with GP infections, GN infections were associated with an increased risk of aseptic loosening. In subsequent animal experiments, LPS, but not LTA, was found to reduce both the number of trabeculae and the bone mineral density. LPS accelerated monocyte differentiation into osteoclast-like cells, whereas LTA did not. Finally, ibudilast—a toll-like receptor (TLR)-4 antagonist—successfully inhibited LPS-induced bone loss and osteoclast activation in mice. PJI, periprosthetic joint infection; GP, Gram-positive; GN, Gram-negative; LPS, lipopolysaccharide; LTA, lipoteichoic acid.

**Table 1 jcm-08-01289-t001:** Analysis of reoperation rates due to aseptic loosening in patients with periprosthetic joint infection (*n* = 320) caused by Gram-positive versus Gram-negative bacteria.

Bacterial PJI	Reoperation Rate %	Reoperation Rate Due to Aseptic Loosening %
GP	25.1% (63/251)	6.3% (4/63)
GN	23.2% (16/69)	31.3% (5/16) *

Abbreviations: PJI, periprosthetic joint infection; GP, Gram-positive; GN, Gram-negative. * *p* = 0.03 (Fisher’s exact test), GN versus GP for PJI. Reoperation rate = reoperation number/total number of patients. Reoperation rate due to aseptic loosening = aseptic loosening number/total reoperation number of patients.

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
