# Peer review of "Periprosthetic Joint Infection Caused by Gram-Positive Versus Gram-Negative Bacteria: Lipopolysaccharide, but not Lipoteichoic Acid, Exerts Adverse Osteoclast-Mediated Effects on the Bone"

_jcm, 2019, doi:10.3390/jcm8091289_

Round 1

Reviewer 1 Report

This is a very interesting study based on the hypothesis that LPS of Gram negatives are responsible for bone loss / bone destruction resulting in a higher rate of aseptic loosening after a PJI due to a Gram negative (compared to Gram positives).

As a clinician I have several questions that need to be addressed and clarified in the manuscript:

Major:

The dose of LPS and LTA used in the study, are these amounts comparable to a clinical situation? Please clarify how this dose was chosen and based on which literature. If it is not clear whether this amount is comparable to a clinical situation, this should be stated as a limitation in the discussion section. Why did the authors not choose to conduct a study in which Gram negative and Gram positive mo's were injected into a bone, treated with antibiotics and then evaluate bone loss? This would be more comparable to the real life situation in patients. Table 1: the way the data are presented are completely unclear. Please delete and replace this in a survival curve in which aseptic failure is depicted as an event at a certain point in time during follow-up. Please take hips and knees together; this would be aseptic loosening during follow-up for Gram negatives in 4/251=1.6% and 5/69=7.2%. It is unclear if these 320 PJIs that were included were all chronic infections and had a revision of the infected implant. Or if acute PJIs were included as well that were treated with DAIR and implant retention. This should be made clear in the result section (this was also completely unclear in the headings of Table 1). Do the authors have data that LPS is still active after infection treatment? How would the authors explain that aseptic loosening occurs after so many years of PJI treatment? If the theory is correct, you would expect immediate problems due to the bone destruction, no?

Minor:

To use the word reimplantation is very confusing in the context of clinicians that work in the field of PJI, because it is generally used in the context of a 2-stage revision where the implant is reimplanted during the 2nd stage. Please just state that you are evaluating the number of aseptic loosenings after revision surgery and antibiotic treatment for a PJI. If data are prospectively collected, you can just state that is was a prospective study. Please remove the word retrospective. 531 patients had a PJI, and 320 were caused by bacteria? What do you mean? That the others were culture negative? Or caused by fungi? Please rephrase, because PJI in general is caused by bacteria. Please add in the material and method section how many mice were used in each arm (and state this also in the result section) There are results depicted on the clinical study, that should be in the result section. Please state as a limitation that only 7.2% of GN PJIs had aseptic loosening during follow-up, this means that if you want to prevent this by treating these patients with a toll-like receptor antagonist, that you over-treat a lot of patients. Is this toll-like receptor antagonist considered as an immunosuppressive drug? And if so, at what time point would you consider to use this in patients? As this can not be used during an active infection. Please comment on this in the discussion section.

Reviewer 2 Report

my review is attached, since copy/paste is obviously not possible

Round 2

Reviewer 2 Report

The reviewer's comments have been adequately answered in the revised version of the manuscript